# A mathematical model for estimating anti-learning when a decision tree solves the parity bit problem

## Abstract

On some data, machine learning displays anti-learning; this means, in the most surprising scenario, that the more examples you place in the training set, the worse the accuracy becomes, until it becomes 0% on the test set. We produce a framework in which this kind of anti-learning can be reproduced and studied theoretically. We deduce a formula estimating anti-learning when decision trees (one of the most important tools of machine learning) solve the parity bit problem (one of the most famously tricky problems of machine learning). Our estimation formula (deduced under certain mathematical assumptions) agrees very well with experimental results (produced on random data without these assumptions).

## 1 Anti-learning, the parity bit problem, and decision trees

Let $f$ be a function. For the purposes of this study, one can assume that the domain $\mathrm{dom} f$ is finite, and the image $\mathrm{im} f = \{0, 1\}$. Split $\mathrm{dom} f$ into two sets, $\mathrm{dom} f = R \cup S$, $R \cap S = \emptyset$. *Machine learning* can be described as the art of using algorithms to predict the values of $f$ on $S$ when the values of $f$ on $R$ are given. In this context, $R$ is called the *training set*, and $S$ is called the *test set*. The percentage of the correct predictions of the values of $f$ on $S$ is called *accuracy*. A random predictor has accuracy 50%. The assumption of machine learning is that after inspecting the values of $f$ on $R$, one can achieve a more than 50% accuracy. In some examples, researchers observed *anti-learning* Kowalczyk & Chapelle (2005); Kowalczyk (2007); Roadknight et al. (2018), that is, after learning on the values of $f$ on $R$, accuracy becomes less than 50%.

It can be useful to say that anti-learning should not be confused with *overfitting*, which is an important but different phenomenon in machine learning. Overfitting can be briefly described as accuracy decreasing towards 50%, whereas anti-learning means that accuracy paradoxically decreases below 50%.

As we started exploring anti-learning, one phenomenon that attracted our attention was that in some scenarios, as we vary the size $|R|$ from slightly more than 0 to slightly less than $|\mathrm{dom} f|$, accuracy monotically decreases from 50% to 0%. For example, Figure 1 shows how accuracy (shown on the vertical axis) changes when we use random forests to predict the parity of a permutation (that is, to predict if a given permutation is odd or even) and vary the size $|R|$ (shown on the horizontal axis as the percentage of the size $|\mathrm{dom} f|$). This paper is our attempt to explain the monotonically decreasing shape of the curve.

To study anti-learning using mathematics, we concentrate on a combination of the parity bit problem and decision trees. Denote the two-element field GF(2) by $F$, and fix a positive integer $n$. The *parity bit* of a vector $(x_1, \ldots, x_n) \in F^n$ is the sum $x_1 + \cdots + x_n$, with the addition performed in $F$. The parity bit is used in many applications; in the context of machine learning, it is an example of a famously hard problem, see Examples 1, 2 below. Exploring what it takes to successfully solve the parity bit problem always leads to fruitful research discussions in machine learning; examples stretch from the 1960s discussion of what perceptrons can calculate the parity bit Minsky & Papert (1969) to the very recent discovery that bees seem to be able to learn parity Howard et al. (2022).

**Example 1.** Let $n = 2$, and let $R = \{(0,0)(1,1)\}$ and $S = \{(0,1)(0,1)\}$. The parity bit of every vector in the training set $R$ is 0, therefore, one predicts that the parity bit of every vector is 0. If we test this prediction on the test set $S$, it is wrong on all vectors of $S$; indeed, the parity bit of every vector in $S$ is 1. Thus, accuracy is 0%.

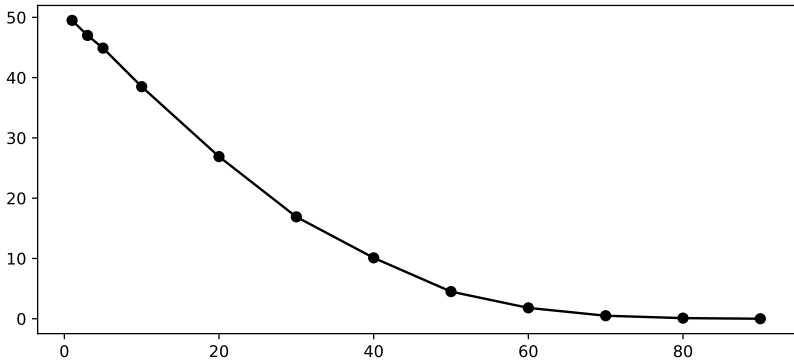

Figure 1: Accuracy when random forests predict permutation parity

**Example 2.** Let $n = 2$, and let $R = \{(0,0)(0,1)\}$ and $S = \{(1,0)(1,1)\}$. The parity bit of every vector $(x_1, x_2)$ in the training set $R$ coincides with $x_2$, therefore, one predicts that the parity bit of every vector is $x_2$. If we test this prediction on the test set $S$, it is wrong on all vectors of $S$; indeed, the parity bit of every vector in $S$ is not $x_2$. Thus, accuracy is 0%.

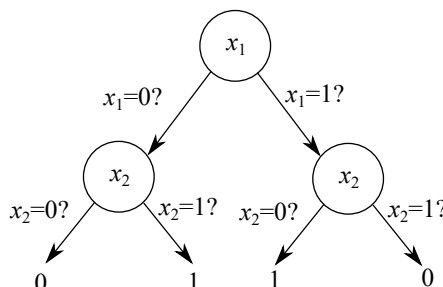

Figure 2: A decision tree calculating the parity bit of $(x_1, x_2)$

Constructions used in Examples 1, 2 are rudimentary examples of a model called a *decision tree*. Figure 2 shows a decision tree calculating the parity bit of $(x_1, x_2)$. The nodes in the tree which are not leaves instruct us to inspect the value of one position in the vector, and the leaves contain a prediction. Note that this tree is intentionally constructed to calculate the parity bit correctly for every vector $(x_1, x_2)$; unlike Examples 1, 2 this tree is not produced by training on a subset $R \subset F^2$. This tree is *balanced*, that is, all paths from the root to a leaf have the same length. By a *leaf set* we will mean the set of vectors defined by the conditions written along one path from the root to a leaf; for example, the rightmost path in the tree defines a leaf set described by equalities $x_1 = 1$, $x_2 = 1$; in this example, the leaf set consists of one vector $\{(1,1)\}$. By the *support* of a leaf set we mean the positions in the vector which feature in the equalities defining the leaf set; for example, in this tree all leaf sets have the same support $x_1, x_2$. The *size* of a tree is defined as the number of its nodes; for example, the trees in Figure 2, Example 1 and Example 2 have sizes $7, 1, 3$.

In this paper we build a theoretical model of anti-learning when a decision tree solves the parity bit problem and compare it with experimental data.

The paper Bengio et al. (2010) also studies performance of decision trees at solving the parity bit problem. Theorem 1 in Bengio et al. (2010) can be re-formulated as stating that accuracy is in the interval between 0% and 50%. Our approach is much more nuanced; our work culminates in producing a formula which estimates an expected value of accuracy as a function of two variables, $|S|$ and the size of the tree.

It should be noted that although the parity bit problem and decision trees are important in themselves in the study of machine learning, our research interests spread far beyond them. As we have said, we first observed

anti-learning resulting in monotonically decreasing accuracy (as shown in Figure 1) when we used random forests to predict the parity of a permutation. We also applied feed-forward neural networks to predict the parity of a permutation; we observed some anti-learning, but results were so noisy and unpredictable that we could not form any conclusions or conjectures. Studying how neural networks cope with predicting the parity of permutations and with similar problems is a subject of our current research. But as to decision trees and random forests, anti-learning reliably shows itself in the shape of monotonically decreasing accuracy, as in Figures 1 and 3. We are certain that the model presented in this paper will help us and other researchers to better understand anti-learning when it occurs in various scenarios.

## 2    Linear-algebraic constructions

By definition, every leaf set is a hyperplane in $F^n$. Denote by $P_0$ (or $P_1$) the set of all vectors in $F^n$ whose parity bit is 0 (or 1). Both $P_0$ and $P_1$ are hyperplanes in $F^n$. These observations suggest that we can use linear algebra to estimate performance of decision trees at solving the parity bit problem. To proceed, we make two assumptions: (A1) we assume that the test set $S$ is a hyperplane, and (A2) we assume that the tree is balanced, therefore, each leaf set has the same dimension. The assumptions (A1) and (A2) are chosen as a delicate compromise which, on the one hand, ensures that our mathematical model covers a wide range of examples and, on the other hand, makes it feasible to use linear algebra and probability theory to estimate accuracy. As we will see in Figure 3, a mathematical model built under these assumptions predicts well accuracy in experiments produced without these assumptions.

From now on, we write vectors as columns. Let the test set $S$ be a hyperplane of $F^n$ defined by a matrix equation $A\mathbf{v} = \mathbf{b}$. The training set is $R = F^n \setminus S$. When we consider an individual leaf set, we denote it by $L$; let $L$ be a hyperplane defined by a matrix equation $C\mathbf{v} = \mathbf{d}$. Let us denote the number of rows in $A$ (in $C$) by $a$ (by $c$); we will assume that the rows of $A$ are linearly independent, and the rows of $C$ are linearly independent. Thus, $c$ is the size of the support of $L$, the dimension of $S$ is $n - a$, and the dimension of $L$ is $n - c$.

Recall that in machine learning, training is performed to maximise the rate of correct predictions on the training set $R$; therefore, for each leaf of the tree, the prediction made by the leaf is chosen to maximise the rate of correct predictions on $R \cap L$, where $L$ is this leaf's leaf set.

Denote by $\mathbf{1}$ a vector in $F^n$ consisting of 1s. Obviously, $P_0$ (or $P_1$) is the set of all vectors $\mathbf{v}$ such that $\mathbf{1}^T\mathbf{v} = 0$ (such that $\mathbf{1}^T\mathbf{v} = 1$).

**Proposition 3.** *The vector $\mathbf{1}^T$ is a linear combination of rows of $A$ and $C$ if and only if $S \cap L$ is a subset of $P_0$ or $P_1$.*

*Proof.* In the notation introduced above, $S \cap L$ is a hyperplane defined by a matrix equation $\begin{bmatrix} A \\ C \end{bmatrix} \mathbf{v} = \begin{bmatrix} \mathbf{b} \\ \mathbf{d} \end{bmatrix}$. Suppose $\mathbf{1}^T$ is a linear combination of rows of $A$ and $C$, that is, for some vector $\mathbf{t} \in F^{a+c}$ we have $\mathbf{t}^T \begin{bmatrix} A \\ C \end{bmatrix} = \mathbf{1}^T$. Hence, for every $\mathbf{v} \in S \cap L$ we have $\mathbf{t}^T \begin{bmatrix} A \\ C \end{bmatrix} \mathbf{v} = \mathbf{t}^T \begin{bmatrix} b \\ d \end{bmatrix}$. As to $\mathbf{t}^T \begin{bmatrix} b \\ d \end{bmatrix}$, it is a scalar, that is, either 0 or 1, and it does not depend on $\mathbf{v}$. Thus, we have either $\mathbf{1}^T\mathbf{v} = 0$ for every $\mathbf{v} \in S \cap L$ or $\mathbf{1}^T\mathbf{v} = 1$ for every $\mathbf{v} \in S \cap L$. Hence, either every $\mathbf{v} \in S \cap L$ is in $P_0$ or every $\mathbf{v} \in S \cap L$ is in $P_1$.

Conversely, suppose that $S \cap L$ is a subset of $P_0$ or $P_1$. Then $\mathbf{1}^T$ is contained in the orthogonal complement of the hyperplane $S \cap L$. The rows of $A$ and $C$ form a spanning set of the orthogonal complement of $S \cap L$. Therefore, $\mathbf{1}^T$ is a linear combination of rows of $A$ and $C$.                                                             □

**Proposition 4.** *If $c < n$ then $L$ contains vectors with both values of the parity bit.*

*Proof.* Consider an arbitrarily chosen vector $\mathbf{v} \in L$. Since $c < n$, there is a position $i$ in $\mathbf{v}$ which is not in the support of $L$. Construct a vector $\mathbf{w}$ which coincides with $\mathbf{v}$ at all positions except $i$, and whose entry at position $i$ is 0 (or 1) if the entry at position $i$ is 1 (or 0) in the vector $\mathbf{v}$. Since $i$ is not in the support of

$L$ and $\mathbf{v} \in L$ and $\mathbf{w}$ coincides with $\mathbf{v}$ at all positions except $i$, we conclude that $\mathbf{w} \in L$. At the same time, since $\mathbf{w}$ and $\mathbf{v}$ differ at exactly one position, we have $\mathbf{1}^{\mathrm{T}}\mathbf{v} \neq \mathbf{1}^{\mathrm{T}}\mathbf{w}$; in other words, one of the two vectors $\mathbf{w}$ and $\mathbf{v}$ is in $P_0$ and the other is in $P_1$. $\qquad\square$

Proposition 4 shows that it is realistic to assume, as we will in Theorem 5, that a leaf set contains vectors with both values of the parity bit. The only exception is $c = n$, when the leaf set contains exactly one vector. This is achieved only if the tree size is unrealistically large, so we assume that $c < n$. (For comparison, in Theorem 1 in Bengio et al. (2010) large unbalanced trees are used, and leaf sets with the support of size $n$ are explicitly considered.)

**Theorem 5.** *Suppose a leaf set $L$ has a non-empty intersection with both the training set $R$ and the test set $S$, and suppose $L$ contains vectors with both values of the parity bit. Then accuracy on $S \cap L$ is either $0\%$ or $50\%$. Namely, it is $0\%$ (it is $50\%$) if $\mathbf{1}^{\mathrm{T}}$ is (is not) a linear combination of rows of $A$ and $C$.*

*Proof.* Each of the subsets $L \cap P_0$ and $L \cap P_1$ is non-empty, and each of them is an $(n - c - 1)$-dimensional hyperplane in $F^n$, therefore, $|L \cap P_0| = |L \cap P_1|$. Suppose $\mathbf{1}^{\mathrm{T}}$ is a linear combination of rows of $A$ and $C$. By Proposition 3, $(S \cap L) \subseteq P_i$, where $i$ is 0 or 1. Hence, $R \cap L$ is a union of two non-overlapping subsets, $L \cap P_{1-i}$ and $(L \cap P_i) \setminus (S \cap L)$. The former subset contains more elements than the latter, therefore, the prediction chosen on this leaf is that every vector in $L$ is in $P_{1-i}$. This prediction is wrong on every vector in $S \cap L$, since $(S \cap L) \subseteq P_i$. Therefore, accuracy on $S \cap L$ is $0\%$.

Now suppose $\mathbf{1}^{\mathrm{T}}$ is not a linear combination of rows of $A$ and $C$. By Proposition 3, $S \cap L$ has a non-empty intersection with both $P_0$ and $P_1$. Thus, $S \cap L \cap P_0$ and $S \cap L \cap P_1$ are hyperplanes in $F^n$ having the same dimension, therefore, $|S \cap L \cap P_0| = |S \cap L \cap P_1|$. Recall that $R = F^n \setminus S$ and $|L \cap P_0| = |L \cap P_1|$; hence, $|S \cap R \cap P_0| = |S \cap R \cap P_1|$. Thus, the same number of elements of $S \cap R$ has parity 0 and parity 1; therefore, the prediction on this leaf will be chosen 0 or 1 at random. Whichever it is, since $|S \cap L \cap P_0| = |S \cap L \cap P_1|$, this prediction will be correct on exactly a half of the vectors in $S \cap L$ and wrong on the other half of the vectors in $S \cap L$. Therefore, accuracy on $S \cap L$ is $50\%$. $\qquad\square$

Theorem 5 enables us to start discussing informally what accuracy we are likely to expect if the test set is small or large. If $a$, the number of rows in $A$, is small then it is 'less likely' that $\mathbf{1}^{\mathrm{T}}$ is a linear combination of rows of $A$ and $C$, but if $a$ is large then it is 'more likely'. Recall that the test set $S$ is an $(n - a)$-dimensional hyperplane of $F^n$, and the training set $R$ is $F^n \setminus S$. Thus, if the training set is small and the test set is large, accuracy is 'likely' to be about $50\%$, whereas if the training set is large and the test set is small, accuracy is 'likely' to be about $0\%$. The next section refines this discussion by producing specific numerical values.

## 3 Estimating accuracy

The following construction will be useful. Let $X$ be a set of size $x$, and let $Y, Z$ be two randomly chosen subsets of $X$ having sizes $y, z$, respectively. Denote the probability of $Y$ and $Z$ having an empty intersection by $\xi(x, y, z)$. From probability theory, $\xi(x, y, z) = \frac{(x-y)!(x-z)!}{x!(x-y-z)!}$, and this value can be easily computed with the help of Stirling's approximation.

To produce an estimation of accuracy, we introduce one more assumption. Let $\langle A \rangle$ and $\langle C \rangle$ denote the spaces (of row vectors) spanned by rows of $A$ and $C$, respectively, where $A$ and $C$ are as defined in the previous section. In other words, $\langle A \rangle$ and $\langle C \rangle$ are the orthogonal complements of $S$ and $L$, respectively. Since we are working with row vectors, for convenience, we shall treat $F^n$ as consisting of row vectors; thus, both $\langle A \rangle$ and $\langle C \rangle$ are subspaces of $F^n$. Our new assumption (A3) is that $\langle A \rangle$ and $\langle C \rangle$ are randomly chosen subsets of $F^n$. This assumption can be described as a scenario in which an algorithm building the decision tree chooses conditions at the nodes of the tree at random. When one considers other classification problems, this assumption would not be justified because the algorithm tries to optimize the tree's performance. However, the parity bit problem, by its construction, is hopelessly hard for a decision tree, therefore, we feel that it is justifiable to assume that the decision tree behaves as if it was chosen at random.

**Proposition 6.** *Under the assumptions of Theorem 5 and (A3), expected accuracy on $S \cap L$ can be estimated as $\frac{1}{2}\xi(2^n, 2^a, 2^c)$.*

*Proof.* Let $p$ be the probability that $\mathbf{1}^{\mathrm{T}}$ is not a linear combination of rows of $A$ and $C$. According to Theorem 5, expected accuracy can be expressed as $0 \cdot (1-p) + \frac{1}{2}p$. In the remaining part of the proof we will show that $p$ can be estimated as $\xi(2^n, 2^a, 2^c)$, hence the result follows.

Recall that we assume that that $\langle A \rangle$ and $\langle C \rangle$ are random subsets of $F^n$. Hence, $\langle C \rangle + \mathbf{1}$ also is a random subset of $F^n$. We can express the fact that $\mathbf{1}$ is not a linear combination of rows of $A$ and $C$ as saying that $\langle A \rangle$ and $\langle C \rangle + \mathbf{1}$ do not intersect. Since $\langle A \rangle$ and $\langle C \rangle + \mathbf{1}$ are two randomly chosen sets of sizes $y = 2^a$ and $z = 2^c$ in $F^n$, whose size is $x = 2^n$, the probability of these sets not intersecting is $\xi(2^n, 2^a, 2^c)$. $\qquad \square$

**Theorem 7.** *Expected accuracy can be estimated as*

$$\frac{1}{2}\xi(2^n, 2^n - 2^{n-a}, 2^{n-c}) + \frac{1}{2}\left(1 - \xi(2^n, 2^n - 2^{n-a}, 2^{n-c})\right) \cdot \xi(2^n, 2^a, 2^c).$$

*Proof.* Consider a randomly chosen element $\mathbf{v}$ of $S$. Since the whole space $F^n$ is split into leaf sets, $\mathbf{v}$ lies in a certain leaf set $L$. We have either $R \cap L = \emptyset$ or $R \cap L \neq \emptyset$. In the former case, due to the absence of any elements of $R$ to base the prediction on, we assume that the prediction for the leaf corresponding to $L$ is made at random, as either 0 or 1. Hence, expected accuracy on $S \cap L$ is 50%. In the latter case, we are in the conditions of Theorem 5, and expected accuracy is as expressed in Proposition 6.

Now estimate the probability that $R \cap L = \emptyset$. For the purposes of this estimation, we treat $R$ as a random subset of $F^n$ of size $x = 2^n - 2^{n-a}$, and $L$ as a random subset of $F^n$ of size $x = 2^{n-c}$. Hence, the probability is estimated as $\xi(2^n, 2^n - 2^{n-a}, 2^{n-c})$. By putting these fragments together, we obtain the formula in the statement of the theorem. $\qquad \square$

To perform comparison with the results of experiments produced on random data, without the linearity assumptions A1, A2 introduced in Section 2, we rewrite the formula in Theorem 7 as shown below, so it does not refer to dimensions of $S$ and leaf sets, but only refers to the size of $S$ and the size of the tree.

**Corollary 8.** *Expected accuracy can be estimated as*

$$\frac{1}{2}\xi(2^n, 2^n - |S|, \frac{2^{n+1}}{t+1}) + \frac{1}{2}\left(1 - \xi(2^n, 2^n - |S|, \frac{2^{n+1}}{t+1})\right) \cdot \xi(2^n, \frac{2^n}{|S|}, \frac{t+1}{2}),$$

*where $t$ is the number of nodes in the tree.*

*Proof.* The formula is produced from the formula in Theorem 7 by substituting $c = \log(t + 1) - 1$ and $a = n - \log|S|$, where log is the binary logarithm. $\qquad \square$

Using the formula in Corollary 8, we compare the estimation of Theorem 7 with the experimental results produced using WEKA machine learning software Witten & Frank (2005). In Figure 3, the horizontal axis is the size of a training set $R$, shown as percentage of the size of $F^n$, $n = 12$. The vertical axis is accuracy. In each experiment, the size $|R|$ is fixed and then $R$ is chosen as a random subset of $F^n$ of this size. Then WEKA is used to produce a decision tree learning on $R$ to predict the parity bit, and then accuracy on $S = F^n \setminus R$ is measured. Note that the tree size is not constant but is, in each experiment, chosen by WEKA's algorithm and varies; we do not show the tree size on this diagram, but you can view all data in the online repository of the data of this paper [the weblink is temporarily removed for the time of reviewing]. Each circle in the line with circles shows accuracy produced in an experiment, and the corresponding diamond in the line with diamonds shows the estimation of accuracy produced by the formula in Corollary 8 based on the same test set size and tree size as in the experiment. What both graphs in Figure 3 suggest is that accuracy monotonically decreases from 50% to 0% as the size of the training set increases. The overall shape of the two graphs is almost the same, so our estimation in Corollary 8 is surprisingly good, taking into account how simple our model is.

In our experiments when we use random forests instead of decision trees, we also observe anti-learning, with approximately the same shape of a graph as in Figure 3 (cf. Figure 1, showing anti-learning when a random forest solves a slightly different parity problem). Of course, a mathematical model attempting to describe anti-learning of an ensemble of decision trees would have to be considerably more complicated than of one decision tree; this is why in this paper we have concentrated on modeling anti-learning of one decision tree.

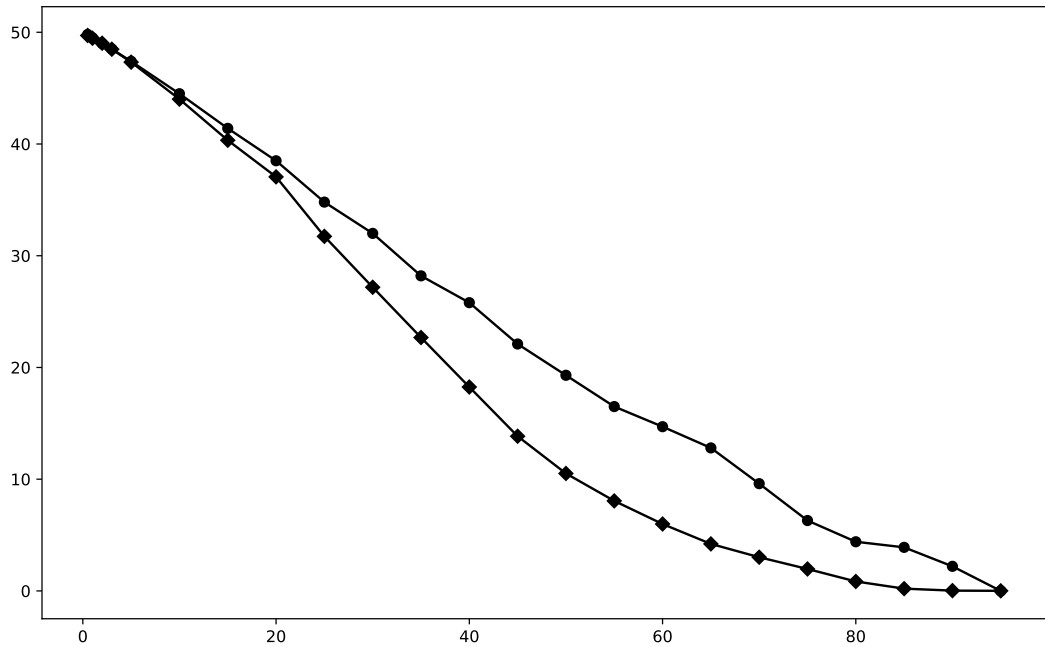

Figure 3: Comparison of accuracy

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
