# OpenReview forum: "A mathematical model for estimating anti-learning when a decision tree solves the parity bit problem"
_TMLR — Rejected by TMLR_

### Review · Reviewer_vNmj · 2022-07-10

**Summary Of Contributions:**

The paper analyzes the decision tree for the parity bit problem and, under a set of assumptions, it demonstrates the anti-learning phenomenon where the test accuracy decreases (below random chance) as the training size gets larger.

**Requested Changes:**

Please address the second point in the weakness section.

**Strengths And Weaknesses:**

**Strength**

- Anti-learning is a counter-intuitive phenomenon at the first glance. In fact, in a similar (but not same) model, the prior work Bengio et al 2010 in its calculations has hinted at this phenomenon. This work analyzes more directly this phenomenon in a clear and systematic way.

- The mathematics is also simple and clean, and the writing is clear.

**Weakness**

- Anti-learning should actually be intuitive if one frames it in the context of a decision tree in the parity bit problem and restricts the test set to only unseen samples. Each leaf in the tree can classify at best at random chance, which is already known from Bengio et al 2010, and therefore, the more training samples are added to tilt this leaf to one side of the decision, the worse the classification on unseen samples has to be. The paper adds several assumptions to help deriving the result in a closed-form formula, but I’m unsure if the paper generates new nontrivial insights.

- The bigger problem I can see is in the set of assumptions that the paper makes use. Firstly the paper assumes that the tree is balanced and assumes that the test set is a hyperplane, i.e. it implicitly assumes a structure on the training set, but the paper does not justify why these assumptions may not collide with each other.

  Secondly the randomness model in the paper is not validated. In Proposition 6, the paper assumes that the test set and each leaf set can be considered as randomly and independently chosen sets, but one ought to be reminded that the leaf set, which is mutually dependent on the training set, cannot be independent from the test set. Likewise in the proof of Theorem 7, the paper assumes that the leaf set and the training set are randomly chosen sets, but again they must be correlated.

---

> ### Author Response · Authors · 2022-07-13
> **thank you, and here is an answer to your question**
>
> Thank you for reading our paper, summarising it and making your thoughtful comments.
>
> You are asking me to explain why we can assume that a leaf set can be treated as a random set for the purpose of calculating probabilities in our paper. It is a very interesting question, and let me explain this. It is possible to imagine many ways in which one can study leaf sets. In particular, each leaf set has what we call a support in our paper. If one is interested in the support of different leaf sets of the same tree, one needs to keep in mind that all the leaf sets of the same tree have supports which overlap with one another. However, in the part of the paper where we deal with probabilities (Section 3) we never compare two leaf sets with one another. We consider an individual leaf set and explore how it interacts with the test set. Of course, different leaf sets can have different orientations relative to the test set, and we believe that we adequately express this fact by using suitable coefficients in Theorem 7. In this context, each leaf set can be treated as a randomly chosen subspace of the right dimension (with the random choice made as explained in the proof of Proposition 6).
>
> More specifically, you are asking why leaf sets can be considered random relative to the test set. The answer to this question lies in the nature of the problem that we consider, namely, the parity bit problem. Each leaf set is a subspace, and there is no way a decision tree will demonstrate a good performance at solving this problem, only perhaps a 'less bad' performance on some individual leaf sets (as we say in Theorem 5). Therefore, if one builds a decision tree at random, or if a clever algorithm (for example, using Gini impurity) builds a decision tree in the context of a given training set, if will not have an effect on the overall performance of the tree. This is why when we solve this particular problem, the parity bit problem, we can assume that leaf sets are randomly chosen.
>
> At this point, you might ask me about how we can locate those individual leaf sets on which Theorem 5 predicts a 'less bad' performance; we do not exactly locate them (because it would be a formidable task), but we predict how many of them we expect to be; we count them in Proposition 6 and Theorem 7.

---

> > ### Comment · Reviewer_vNmj · 2022-08-24
> > **reply**
> >
> > Thanks for the clarification.
> >
> > Consider the following formalization. We draw a test set $S$ at random such that it is a hyperplane (i.e. given a fixed size, $S$ is a uniform random variable on the set of all subsets of that size of $F^n$, conditional on that these subsets are hyperplanes), and the training set $R$ is the complement of $S$. A decision tree is an algorithm $\cal{A}$, which takes $R$ as input and so can be viewed as a functional of $S$. Let $\cal{L}$ be the set of leaf sets of $\cal{A}(S)$, and $L$ be some leaf set in $\cal{L}$ (perhaps drawn randomly from $\cal{L}$). To establish the results in the paper, one must establish the properties of the joint distribution between $L$ and $S$, conditioning on that $\cal{A}(S)$ gives a balanced tree, as these properties are used in the proofs of the results.
> >
> > What I'm looking for is a proof of those properties.

---

### Review · Reviewer_7J45 · 2022-08-19

**Summary Of Contributions:**

The paper talks about Anti Learning in Decision Tress for parity bit problem. Anti-learning is the situation when your model will give consistently worse of accuracy (i.e. accuracy less than 50% for a two class problem) on unseen data, no matter how hard you optimize your algorithms. The paper provides proof on why if the model is trained on more data, it might mean a larger drop in accuracy for parity bit problem in Decision Trees. The following helps differentiate between Anti-Learning and Overfitting.
The paper describes Parity Bit Problem and moves into understanding Training, Testing and Leaf Set as Hyperplanes. The hyperplanes are described as Matrix equations. Theorem Five explains how accuracy can be 0%(or 50%) on test data. The theorem also explains cases where Data size increases but testing accuracy goes to 0%. The next section using Stirling’s approximation to provide an estimate of expected accuracy.


**Broader Impact Concerns:**

The paper discusses a framework to estimate Anti-Learning and generate datasets to study Anti-Learning

**Requested Changes:**

The following recommendations will strengthen the work in my view

The paper makes two assumptions A1 and A2. It would have been great if the assumption on balanced tree could be explained further.
For proposition 2, it would be great if the writer could add examples to explain the Proposition in detail. Similarly for Theorem 5 it would be nice if more justification was added for c-1 dimension for P0 and P1 Subsets with L.


**Strengths And Weaknesses:**

Strength
The paper does a good job in Formalizing Training, Testing and Leaf Set Hyperplane space.
Weakness
The paper makes two assumptions A1 and A2. It would have been great if the assumption on balanced tree could be explained further.
For proposition 2, it would be great if the writer could add examples to explain the Proposition in detail. Similarly for Theorem 5 it would be nice if more justification was added for c-1 dimension for P0 and P1 Subsets with L.

---

### Review · Reviewer_MJYL · 2022-08-21

**Summary Of Contributions:**

This paper focuses on Anti-learning, which is a phenomenon where a trained model is less than 50% accurate on unseen data. The authors focus on the manifestation of anti-learning in decision tree models built to solve the parity bit problem. The authors exploit the structure of a decision tree to pose the leaf sets as hyperplanes, whose parities are 0 or 1. This enables the use of matrix algebra to arrive at a formal estimation for the expected accuracy of such a model on unseen data.

**Requested Changes:**

1. Any intuition on how these arguments hold for popular ensembles of decision trees (like random forests, boosted trees etc.) would be great.


**Strengths And Weaknesses:**

1.  Clearly differentiates anti-learning from overfitting
2.  Good use of linear & matrix algebra notations to formalize arguments

---

### Decision · Action_Editors · 2022-10-08

**Recommendation:** Reject

**Comment:**

The main issue with the paper is that the final proof seems to be sketchy and relied on implicit assumptions, to be discovered by the reviewers. The authors revised that and added further assumptions. The original assumptions were already pointed out to be a weakness of the paper, as there was not enough discussion on those.



**Audience:**

Anti-Learning audience.

**Claims And Evidence:**

Some of the results in the authors' proof relied on statements that were found to be ill-justified. The authors revised these statements and reformulated them as assumptions, which greatly weakened the final statement.